Pediatrics HIV-positive status disclosure and its predictors in Ethiopia: a systematic review and meta-analysis

Belay Getaneh Mulualem 1
Yehualashet Fikadu Ambaw 2
Ewunetie Amare Wondim 1
Atalell Kendalem Asmare kedasmar@gmail.com 1
1 Department of Pediatrics and Child Health Nursing, School of Nursing, College of Medicine and Health Sciences, University of Gondar , Gondar , Amhara , Ethiopia
2 Department of Community Health Nursing, School of Nursing, College of Medicine and Health Sciences, University of Gondar , Gondar , Amhara , Ethiopia
Mora-Montes Héctor
Electronic publication date: 2022 Aug 23
Publication date: 2022
Volume: 10
Electronic Location ID: e13896
Received 2022 May 24; Accepted 2022 Jul 22
Copyright: ©2022 Belay et al.
Copyright year: 2022
Copyright holder: Belay et al.
License: This is an open access article distributed under the terms of the Creative Commons Attribution License, which permits unrestricted use, distribution, reproduction and adaptation in any medium and for any purpose provided that it is properly attributed. For attribution, the original author(s), title, publication source (PeerJ) and either DOI or URL of the article must be cited.
License URL: https://creativecommons.org/licenses/by/4.0/

Keywords: Children, Disclosure status, Ethiopia, HIV/AIDS, Antiretroviral therapy

Funding: The author received no funding for this work.

==============================
Introduction

HIV-positive status disclosure for children is challenging for family members, guardians, and healthcare professionals. Disclosure is very challenging, particularly for children, yet no systematic synthesis of evidence accurately measures HIV-positive status disclosure in children. This systematic review and meta-analysis study aimed to quantify the national prevalence of pediatric HIV-positive status disclosure in Ethiopia and identify factors associated with HIV-positive status disclosure.

Method

We systematically searched PubMed, EMBASE, Web of Science databases, and google scholar for relevant published studies. Studies published in the English language and conducted with cohort, case-control, and cross-sectional designs were eligible for the review. The primary and secondary outcomes of the study were HIV-positive status disclosure and factors associated with HIV-positive status disclosure, respectively. The quality of the included studies was assessed using the Joanna Briggs Institute critical appraisal tools. A random effect- model was used to estimate the pooled prevalence of HIV-positive status disclosure. Heterogeneity and publication bias of included studies was determined using I2 and Egger’s test, respectively.

Result

From 601 records screened, nine relevant studies consisting of 2,442 HIV-positive children were included in the analysis. The overall pooled prevalence of HIV-positive status disclosure among children living with HIV/AIDS in Ethiopia was 31.2% (95% CI [23.9–38.5]). HIV-negative status of caregivers (AOR: 2.01; 95% CI [1.28–3.18]), long duration on ART (greater than 5 years) (AOR: 3.2; 95% CI [1.77–5.78]) and older age of the child (>10 years) (AOR: 7.2; 95% CI [4.37–11.88]) were significantly associated with HIV-positive status disclosure.

Conclusion

Low prevalence of pediatric HIV-positive status disclosure was observed in Ethiopia. The longer duration of ART, the HIV-negative status of the caregiver, and older age greater than 10 years were the predictors of pediatric HIV-positive status disclosure. Health system leaders and policymakers shall design training and counseling programs for healthcare professionals and caregivers to enhance their awareness about HIV-positive status disclosure.

Trial registration

This review was registered under PROSPERO and received a unique registration number, CRD42019119049.

Introduction

An estimated 2.8 million children were living with Human Immune Deficiency Virus (HIV) at the end of 2020. Of which majority live in Sub-Saharan Africa (Tadesse, Foster & Berhan, 2015; Gachanja & Burkholder, 2016; Sibhatu Biadgilign, Amberbir & Deribe, 2009). The initiation and availability of cost-effective antiretroviral therapy (ART) and quality patient care significantly improved the health of children living with HIV. As a result, more children living with HIV are growing into adolescence and adulthood (Sariah et al., 2016; Natapakwa Skunodom et al., 2006). HIV-positive status disclosure is an emotional practice for parents and their children (Gachanja & Burkholder, 2016). Even though most pediatric HIV infections occur vertically, infected children are unaware of their serostatus for a long time (Ubesieab et al., 2016). The World Health Organization (WHO) recommends that children should be informed of their HIV status at ages 6 to 12 years and full disclosure of HIV and AIDS be offered in a caring and supportive manner at about 8 to 10 years (World Health Organization, 2011b).

Disclosing HIV-positive status for children is very challenging and has to be done cautiously, to minimize stigma, harmful reactions, family and community disintegration, and psychological disruption (Tadesse, Foster & Berhan, 2015; Sariah et al., 2016; Machtinger et al., 2015). It needs parents’ acknowledgment, cooperation and willingness, and the child’s cognitive and emotional ripeness (Negese et al., 2012).

Previous studies argued that HIV-positive status disclosure enhances ART adherence by creating awareness and a better understanding of the disease and the relevance of drug adherence (Vaz et al., 2011; Paul Motshome, 2014). Moreover, delaying disclosure might result in unplanned disclosure through media, school, or the communities and leads to emotionally hostile and withdrawal (Madiba & Mokwena, 2012).

Despite the promising importance of HIV status disclosure, when and how to inform the status of children remains challenging and needs extensive research. On the other hand, fear of violence, desertion or blame, stigma, and discrimination are identified as some barriers to disclosing children’s HIV-positive status (Natapakwa Skunodom et al., 2006). Health care professionals in collaboration with caregivers should develop strategies on how and when to disclose the status of a child at a suitable age and approach (Negese et al., 2012; Johanna & Mahloko, 2012). However, there is no systematic synthesis of evidence that accurately measures HIV-positive status disclosure in children. This systematic review and meta-analysis study aimed to quantify the national prevalence of pediatric HIV-positive status disclosure and identify factors associated with HIV-positive status disclosure in Ethiopia, to come up with evidence to improve HIV status disclosure in children.

Methods

We conducted a comprehensive systematic review and meta-analysis approach following the Preferred Reporting Items of Systematic Reviews and Meta-analysis (PRISMA). The review protocol was registered in the International Prospective Register of Systematic Reviews (PROSPERO) CRD42019119049.

Inclusion criteria

All observational studies (cohort, case-control, and cross-sectional) reporting the prevalence of pediatric HIV status disclosure and its predictors were included in this systematic review and meta-analysis. Our search was limited to studies conducted in Ethiopia and published in the English language.

Exclusion criteria

Systematic reviews and meta-analyses studies, conference papers, editorials, trials, articles without full texts, unpublished studies, and qualitative studies were excluded from this review.

Outcome measures

Pediatrics HIV-positive status disclosure Children aged between 6 and 15 years who are aware of their HIV-positive status (FMOH, 2017).

Partial disclosure Sharing information to the child about his/her illness but not using the word “HIV” (FMOH, 2017).

Full disclosure Telling the child that he/she is living with HIV (FMOH, 2017).

Databases and searching strategy

We systematically searched PubMed, EMBASE, web of science, and google scholar databases for relevant studies. Additionally, a hand-searching of reference list of studies was done. We retrieved studies from databases using the following searching terms, “disclosure”, “HIV positive”, “HIV positive status disclosure”, “disclosure status”, “HIV status”, “caregiver”, “children”, “pediatrics”, “HIV/AIDS”, “Antiretroviral therapy”, “age of the child”, “seropositive status”, “prevalence”, “predictors”, “factors”, “risk factors”, and “Ethiopia”. The searching string was developed using “AND” and “OR” BOOLEAN operators.

Study selection

GMB imported all retrieved articles to Endnote version 7 citation manager and removed duplicated studies. Title and full-text screening were done by two independent authors (GMB and FAY). The full-text review was also conducted by GMB and KAA to decide the final eligibility of the articles. Disagreements were resolved through discussion.

Quality assessment

The quality of included studies was assessed by two authors (GMB and AW) using the Joanna Briggs Institute (JBI) quality appraisal tool (JB Institute, 2017). Studies were considered low risk or good quality whenever fitted to 50% and above to the JBI quality assessment criteria. A thorough review of papers, discussions, consensus, and repetition of quality assessment was done to manage disagreements regarding quality assessment.

Data extraction

Data were extracted from the included articles using piloted data extraction form. Three independent authors (GMB, KAA, and FAY) extracted the data on piloted data extraction form. We collected the author’s name, year of publication, study area, region, population, study design, sample size, and the prevalence of HIV-positive status disclosure. A separate Excel spreadsheet was prepared to identify the most prevailing factors that demonstrate a significant association with HIV-Positive status.

Data analysis

The pooled prevalence of pediatric HIV-positive status disclosure was determined by a weighted inverse variance random-effects model (Borenstein et al., 2010), using STATA version 11. Regional variation of the studies was assessed using subgroup analysis. The percentage of total variation between studies due to heterogeneity was assessed by I2 (Higgins et al., 2003). Heterogeneity was considered low, moderate, or high when I2 values were below 25%, between 25% and 75%, and above 75%, respectively (Higgins et al., 2003). Publication bias was assessed by funnel plot and Egger’s test. P-values less than 0.05 were considered statistically significant.

Result

Searching results

We retrieved 652 articles from different sources (569 from PubMed, 54 from Google scholar, 12 from EMBASE, 11 from the web of science, and 6 from reference lists of the included studies). After the removal of duplicates, 601 unique records were screened by title and abstracts, which resulted in 17 potential articles for full-text review. Based on the full-text review, eight articles were excluded from the review. Finally, nine articles were included in this systematic review and meta-analysis to estimate the overall pooled prevalence of HIV-positive status disclosure and to examine the impact of child age, duration on ART, and HIV status of the caregiver on the HIV-positive status disclosure in Ethiopia (Fig. 1).

Figure 1 Flow diagram of articles selection and screening process.

Characteristics of the included studies

In this systematic review and meta-analysis, a total of 9 studies with 2,442 participants were included. Out of the nine studies, three (Abebe & Teferra, 2012; Argaw & Gedlu, 2016; Biadgilign et al., 2011) were from Addis Abeba, three (Alemu, Berhanu & Emishaw, 2013; Negese et al., 2012; Tamir, Aychiluhem & Jara, 2015) from the Amhara region, one (Lencha et al., 2018) from Oromia, one from (Tadesse, Foster & Berhan, 2015) SNNPR, and one (Mengesha, Dessie & Roba, 2018) from Dire Dawa. All included studies (Tadesse, Foster & Berhan, 2015; Abebe & Teferra, 2012; Argaw & Gedlu, 2016; Biadgilign et al., 2011; Alemu, Berhanu & Emishaw, 2013; Negese et al., 2012; Tamir, Aychiluhem & Jara, 2015; Lencha et al., 2018; Mengesha, Dessie & Roba, 2018) were done with a cross-sectional study design. The highest prevalence was reported in a study in Dire Dawa (49.4%) (Mengesha, Dessie & Roba, 2018) and the lowest in Addis Abeba (16.3%) (Abebe & Teferra, 2012) (Table 1).

Quality of included studies

The quality of all included studies was suitable for review based on JBI critical appraisal quality assessment tool result (Table 1).

Meta-analysis

The funnel plot showed a symmetrical alignment of the studies (Fig. 2), which indicate no significant publication bias and the Egger’s regression test also confirms the absence of publication bias (P = 0.203) because the p-value is greater than 0.05.

Table 1 Characteristics of included studies that report pediatrics HIV-positive status disclosure.

Author	Year of publications	Study area	Region	Study design	Study population	Sample size	Prevalence (%)	Quality	
Alemu A et al.	2013	Bahir Dar	Amhara	cross sectional	children 6–14 years on ART	231	31.5	Low risk	
Argaw T et al.	2016	Addis Ababa	Addis Ababa	cross sectional	6–18 years of age	233	32	Low risk	
Biadglign S et al.	2011	Addis Ababa	Addis Ababa	cross sectional	children on ART	390	17.4	Low risk	
Tamir Y et al.	2014	East gojjam	Amhara	cross sectional	6–15 years of age	300	33.3	Low risk	
Mengesha MM et al.	2018	Dire Dawa	Dire Dawa	cross sectional	child- mother pairs	310	49.4	Low risk	
Negese D et al.	2012	Gondar	Amhara	cross sectional	5–15 years of age	428	39.5	Low risk	
Tadese BT et al.	2015	Hawassa	SNNPR	cross sectional	5–18 Years of age	177	33.3	Low risk	
Abebe W et al.	2012	Addis Ababa	Addis Ababa	cross sectional	school age children	173	16.3	Low risk	
Lencha B et al.	2018	Oromia	Oromia	cross sectional	school age children	200	28.5	Low risk	
Notes.

N.B; SNNR, Southern Nations, Nationalities and Peoples Region.

Figure 2 Funnel plot with pseudo 95% CI for publication bias with Selogp in the y-axis and logp in the x-axis.

Prevalence of HIV-positive status disclosure

A total of 9 studies with 2,442 participants were analyzed in this meta-analysis to estimate the pooled prevalence of pediatric HIV-positive status disclosure in Ethiopia. Consequently, the overall pooled prevalence of pediatrics HIV-positive status disclosure in Ethiopia was 31.2% (95% CI [23.9–38.5%]; I2 = 94%; P = 0.000) (Fig. 3).

Figure 3 Forest plot for the prevalence of HIV-positive status disclosure with 95% CI.

The midpoint and the length of each segment indicate the prevalence and 95% CI of each study respectively. The diamond shape reveals the pooled prevalence of HIV-positive status.

Sub-group analysis was conducted by Amhara region and Addis Ababa city only due to the lack of adequate studies in other regions. Hence, the pooled prevalence of pediatrics HIV positive status disclosure were 35.03% (95% CI [30.09–39.98]; I2 = 61.8%; P = 0.073) and 21.72% (95% CI [12.78–30.67]; I2 = 89.5%; P = 0.000) in Amhara region and Addis Ababa city respectively (Fig. 4).

Figure 4 Forest plot for the subgroup analysis by region with 95% CI.

The midpoint and length of each segment reveal the prevalence and 95% CI of each study respectively. The diamond shape shows the pooled prevalence of HIV-positive status disclosure.

Predictors of HIV-positive status disclosure

A total of eight studies (Tadesse, Foster & Berhan, 2015; Abebe & Teferra, 2012; Argaw & Gedlu, 2016; Biadgilign et al., 2011; Alemu, Berhanu & Emishaw, 2013; Negese et al., 2012; Tamir, Aychiluhem & Jara, 2015; Lencha et al., 2018; Mengesha, Dessie & Roba, 2018) reported extractable data, to determine the association between the child’s age and pediatric HIV-positive status disclosure. Consequently, the pooled OR of pediatrics HIV positive status disclosure for those greater than 10 years of age was found to be 7.2 (95% CI [4.37–11.88]; I2 = 78.8%; P = 0.000). Hence, children above the age of 10 years were 7.2 times more likely to be disclosed their HIV-positive status as compared to their counterparts (Fig. 5).

Figure 5 Forest plot for OR of HIV positive status disclosure among children above the age of 10 years.

The midpoint and the length of each segment reveal OR and 95% CI of each study. The diamond shape shows the pooled OR.

Four studies (Abebe & Teferra, 2012; Alemu, Berhanu & Emishaw, 2013; Negese et al., 2012; Lencha et al., 2018) were pooled to assess the association between HIV status of caregivers and pediatric HIV-positive status disclosure. As a result, children whose caregivers are HIV negative were nearly 2 times more likely to be informed about their HIV as compared to their counterparts (OR 2.01(95% CI [1.28–3.18]; I2 = 48.7%; P = 0.119) (Fig. 6).

Figure 6 Forest plot for OR of HIV positive status disclosure among HIV negative caregivers.

The midpoint and length of each segment indicate the OR and 95% CI respectively. The diamond shape reveals the pooled OR.

Out of 9 studies, five studies (Tadesse, Foster & Berhan, 2015; Argaw & Gedlu, 2016; Alemu, Berhanu & Emishaw, 2013; Tamir, Aychiluhem & Jara, 2015; Lencha et al., 2018) showed the association between duration of stay on ART and HIV-positive status disclosure. The overall pooled odds ratio of HIV-positive status disclosure among children who stayed on ART for five years and more were 3.2 times more likely to be informed of their HIV status than those who stayed less than 5 years on ART (3.2 (95% CI [1.77–5.78]; I2 = 70.5%; P = 0.009) (Fig. 7).

Figure 7 Forest plot for OR of HIV positive disclosure status among children more than 5 years with 95% CI.

The midpoint and the length of the segment show the OR and 95% CI of each study. The diamond shape shows the pooled OR.

Discussion

Despite the progressive achievements in diagnosing and managing HIV /AIDS, disclosing HIV-positive status to children remains a challenge for parents, caregivers, and health professionals. This systemic review and meta-analysis aimed to assess the pooled estimate of pediatric HIV-positive status disclosure and its predictors in Ethiopia. The age of the child (six years and above), cognitive and emotional development, and health status of the child were considered to disclose the HIV status of children in Ethiopia (FMOH, 2017).

The overall pooled prevalence of pediatric HIV-positive status disclosure was 31.2% (95% CI [23.9–38.5]) in Ethiopia, which is higher than studies conducted in Kenya (11.1%) (Turissini et al., 2013), Ghana (21%) (Kallem et al., 2011), and Nigeria (30.9%) (Odiachi & Abegunde, 2016), but lower than studies conducted in Rwanda (65%) (Odiachi & Abegunde, 2016). The difference might be due to variations in HIV-related policies, health system programs, and population cultural practices.

According to the WHO recommendation, school-aged children with cognitive and emotional maturity should be disclosed their HIV-positive status (World Health Organization, 2011a). However, many children didn’t know their HIV status until they become adolescents (Vreeman et al., 2013).

Inconsistent with previous studies conducted in Malawi (Chilemba CPaE, 2015), Rwanda, Nigeria (Odiachi & Abegunde, 2016), and the USA (Marıa, Consuelo & Malow, 2013) children aged 10 years and above were seven times more likely to be disclosed than their counterparts. This could be because older children repeatedly ask their families why they are taking medication. Furthermore, older children may learn and hear about HIV from books, friends, and school teachers which could increase their awareness about HIV/AIDS.

Duration of stay on ART was another predictor associated with disclosure of HIV status in children. Those children who were on ART for five years and above were three times more likely to be informed of their HIV status than their counterparts. This finding was supported by a systematic review in resource-limited settings (Vreeman et al., 2013). This could be because children who stayed a long time on ART would try to investigate, what is going on with them and ask the healthcare professionals and their families. The other reason might be linked with the age of the children, months on ART linearly increased with child age.

This review identified the HIV status of the caregivers as a predictor of disclosure. Those caregivers with HIV-negative status were two times more likely to disclose children’s HIV status than HIV-positive ones. This finding is in line with a study conducted in Uganda (Ssali et al., 2010), India (Bhattacharya, Dubey & Sharma, 2010), and Columbia University (Wiener et al., 2007). The possible explanation might be due to the fear of blame and accountability for the child’s illness in the case of HIV-positive caregivers.

This systematic review and meta-analysis had paramount importance to come up with evidence to suggest policymakers, health professionals, and families. Disclosing the status at the appropriate time and place with the appropriate personnel is very crucial to improving drug adherence, minimizing the social stigma, and preventing accidental disclosure through media or school. However, this study has some limitations, firstly, the search strategy was limited to papers published in the English language. The second limitation of this study is the lack of studies in some regions of the country.

Conclusion

The overall prevalence of pediatric HIV-positive status disclosure was found to be low in Ethiopia. The longer duration of ART, negative HIV status of caregivers, and older age of children were the predictors of HIV-positive status disclosure among children in Ethiopia.

Thus, the Ethiopian Ministry of Health, HIV-related program managers, and stakeholders should design strategies for timely disclosure of children through the collaboration of caregivers and health professionals.

Supplemental Information

Supplemental Information 1 PRISMA checklist

Click here for additional data file.

Supplemental Information 2 Reason for this systematic review and meta-analysis

Click here for additional data file.

We acknowledge the authors of the primary study included in this study.

Abbreviation

AIDS Acquired Immunodeficiency Virus

ART Antiretroviral Therapy

AOR Adjusted Odds Ratio

CI Confidence Interval

HIV Human Immunodeficiency Virus

JBI Joanna Briggs Institute

OR Odds Ratio

SNNPRS South Nation, Nationalities People Regional State

Additional Information and Declarations

Competing Interests

Author Contributions

Data Availability

The authors declare there are no competing interests.

Getaneh Mulualem Belay conceived and designed the experiments, performed the experiments, analyzed the data, prepared figures and/or tables, authored or reviewed drafts of the article, and approved the final draft.

Fikadu Ambaw Yehualashet conceived and designed the experiments, performed the experiments, authored or reviewed drafts of the article, and approved the final draft.

Amare Wondim Ewunetie conceived and designed the experiments, performed the experiments, analyzed the data, authored or reviewed drafts of the article, and approved the final draft.

Kendalem Asmare Atalell conceived and designed the experiments, performed the experiments, analyzed the data, prepared figures and/or tables, authored or reviewed drafts of the article, and approved the final draft.

The following information was supplied regarding data availability:

Characteristics of included studies are available in Table 1.

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
