# Peer review of "Pediatrics HIV-positive status disclosure and its predictors in Ethiopia: a systematic review and meta-analysis"

_PeerJ, doi:10.7717/peerj.13896_

## Round 0.1 · original submission · Minor Revisions

Two experts assessed your manuscript and are positive about the publication of it after addressing the minor points raised.

Reviewer 1 ·

Basic reporting

Language and grammar
The English language and grammar could be improved. There are some orthographical and redaction errors. Some examples include lines 10, 24, 31, 134, but the whole text should be reviewed to ensure the correct writing.

Literature and background
The information provided in the background is brief and concrete, which makes understanding simple.
Some comments on grammar and language are mentioned on the PDF.

Structure, figures, and tables
The structure of the article is good and organized. However, figures are disorganized and appear to have low quality. This should be corrected, especially in Figure 1.

Experimental design

Research question
The research question is well defined, discussed and concluded.

Rigorous investigation
The approaches and methodologies used are appropriate to answer the initial question. However, I do wonder if 9 articles are enough to have statistically robust results.

Methods
Methodology is appropriate and understandable.

Validity of the findings

I consider data is well validated and conclusions are in accordance to what was done in the paper.

Additional comments

In general, the research question is interesting, and the methodologies used are appropriate to answer it, also, the results are well interpreted in the discussions.
There are some questions regarding the discussion in the PDF file, that might complement the information already given.

·

Basic reporting

It is a work of great relevance in the area of ​​epidemiology in pediatric HIV. It highlights the importance of knowledge about the seropositivity of children and their adherence to antiviral treatment. In addition to the fact that it is beneficial for children to know their condition in order to integrate it into their way of life. It the importance of keeping the population informed about this viral infection, the caregivers and health personnel, and the family, but with enough information to avoid discrimination and family disintegration. Children's cognitive ability and maturity should be taken into account in disclosing their seropositive status. This study reveals that it is of great benefit for adherence to antiviral treatment that the child knows their status, and that in Ethiopia the percentage of seropositive children who know their seropositivity is low compared to other countries. The predictors to determine whether the child is aware or not were that the caregiver was seronegative and that the child was older than 10 years.

Experimental design

No comment

Validity of the findings

The statistical tests applied to the meta-analysis were adequate, so the review has statistical validity.

Additional comments

To improve the clarity of the graphs, I did not find the graph that shows the data that shows that the seronegative caregiver is a predictor. They could include it, as well as a better-formatted table to make it more accessible to the reader. Improve the clarity of graphics. I did not find the graph of the data that shows that the seronegative caregiver is a predictor. They could include it, as well as a better-formatted table to make it more accessible to the reader. It could improve the discussion regarding other similar studies in other countries, and the impact of knowledge of seropositive status in children. Can you explain why Egger's test for your analysis? Give more emphasis to the relevance of her work and a possible strategy for children to be aware of their seropositivity at earlier ages.

---

## Round 0.2 · accepted · Accept

The manuscript was significantly improved after addressing all the Reviewers' comments. As a consequence, it is now suitable for publication in PeerJ.